# How Reversible Are the Effects of Fumed Silica on Macrophages? A Proteomics-Informed View

**DOI:** 10.3390/nano10101939

**Published:** 2020-09-29

**Authors:** Anaelle Torres, Bastien Dalzon, Véronique Collin-Faure, Hélène Diemer, Daphna Fenel, Guy Schoehn, Sarah Cianférani, Marie Carrière, Thierry Rabilloud

**Affiliations:** 1Laboratory of Chemistry and Biology of Metals, University Grenoble Alpes, CNRS, CEA, UMR 5249, CEDEX 09, 38054 Grenoble, France; anaelle.torres@cea.fr (A.T.); bastien.dalzon@cea.fr (B.D.); veronique.collin@cea.fr (V.C.-F.); 2Laboratoire de Spectrométrie de Masse BioOrganique, Université de Strasbourg, CNRS, IPHC UMR 7178, F-67087 Strasbourg, France; hdiemer@unistra.fr (H.D.); sarah.cianferani@unistra.fr (S.C.); 3Institute for Structural Biology, Université Grenoble Alpes, CNRS, CEA, F-38000 Grenoble, France; daphna.fenel@ibs.fr (D.F.); guy.schoehn@ibs.fr (G.S.); 4Chimie Interface Biologie pour l’Environnement, la Santé et la Toxicologie (CIBEST), University Grenoble-Alpes, CEA, CNRS UMR 5819, IRIG-SyMMES, F-38054 Grenoble, France; marie.carriere@cea.fr

**Keywords:** amorphous silica, pyrolytic silica, macrophages, inflammation, persistence, proteomics

## Abstract

Synthetic amorphous silica is one of the most used nanomaterials, and numerous toxicological studies have studied its effects. Most of these studies have used an acute exposure mode to investigate the effects immediately after exposure. However, this exposure modality does not allow the investigation of the persistence of the effects, which is a crucial aspect of silica toxicology, as exemplified by crystalline silica. In this paper, we extended the investigations by studying not only the responses immediately after exposure but also after a 72 h post-exposure recovery phase. We used a pyrolytic silica as the test nanomaterial, as this variant of synthetic amorphous silica has been shown to induce a more persistent inflammation in vivo than precipitated silica. To investigate macrophage responses to pyrolytic silica, we used a combination of proteomics and targeted experiments, which allowed us to show that most of the cellular functions that were altered immediately after exposure to pyrolytic silica at a subtoxic dose, such as energy metabolism and cell morphology, returned to normal at the end of the recovery period. However, some alterations, such as the inflammatory responses and some aldehyde detoxification proteins, were persistent. At the proteomic level, other alterations, such as proteins implicated in the endosomal/lysosomal pathway, were also persistent but resulted in normal function, thus suggesting cellular adaptation.

## 1. Introduction

Nanomaterials based on amorphous silica are widely used in the industry for various purposes, e.g., as abrasives or as mineral charges in consumer products such as tires. This, in turn, brings the question of their toxicological effects. While crystalline silica is known to be the etiological agent of silicosis through a permanent inflammation [1], amorphous silica has been shown to induce transient inflammation in vivo and in vitro [2], though this inflammation can also be pronounced [3].

This strong difference in the toxicity of amorphous vs. crystalline silica has brought many questions about the mechanisms at play, and it has been shown that crystallinity and, especially, the presence of sharp edges are critical determinants for the persistence of the toxic effects in crystalline silica [4,5]. The fact that persistent effects can be obtained with some types of amorphous silica [5] raises the question of the relative persistence of effects for the different types of amorphous silica that are currently in use. In this framework, it is interesting to compare the effects of amorphous silicas produced by wet or pyrolytic routes, as they show different surface chemistries [6].

Pyrolytic silica has been shown to induce stronger effects on cells than amorphous silica synthetized by a wet route [6,7,8], and this effect has been correlated with surface defects in silica [7], a phenomenon that has also shown to be at play with vitreous [5] and crystalline [4] silicas.

However, studies about the effects of pyrolytic silica on cells have all been conducted to date in an acute exposure mode, where the biological endpoints are read immediately after exposure to the silica particles for times ranging from 6 to 48 h [7,8]. While such studies are very informative in documenting the intensity and nature of the cellular responses, they cannot be used to predict the persistence of the effects of the silica particles over time. Moreover, these studies have investigated a few parameters at a time, e.g., the secretion of nitric oxide or pro-inflammatory cytokines, which are relevant parameters in the frame of chronic inflammation but do not describe the changes occurring in cell physiology at a broader scale that may influence cellular responses in complex environments. Wider studies such as those based on omics techniques can be useful to address such questions. Transcriptomic studies of the effects of silica on cells have been published and have shown unexpected results, such as a modulation of the expression of genes implicated in the epithelial–mesenchymal transition or in cell adhesion [9]. Proteomic studies have also been applied to silica nanoparticle toxicology on several cell types such as lung epithelial cells [10] and macrophages [11]. Here again, unexpected effects have been predicted from omics studies and verified by targeted studies, e.g., a cross-toxicity between colloidal silica nanoparticles and DNA alkylating agents such as styrene oxide [11]. It thus seemed interesting to apply omics studies in a cell culture system that allowed for the investigation of the persistence of the effects of silica nanoparticles. One system consists of exposing the cells to the nanomaterial of interest and then removing the exposure medium and replacing it with a fresh culture medium in order to investigate the cellular responses during this recovery period. This type of exposure has been applied in the case of silver nanowires [12], and it has shown a persistence of the inflammatory effects induced by long nanowires, as is consistent with the effects observed in an in vivo exposure study [13].

In the present study, we thus combined the recovery in vitro system with combined proteomic and targeted studies to investigate the persistence of the effects of pyrolytic silica on macrophages.

## 2. Materials and Methods

Most experiments have been essentially performed as described in previous publications [11,14,15,16]. Details are given here for the sake of the consistency of the paper. All experiments were performed on independent biological triplicates.

### 2.1. Nanoparticles

The pyrolytic silica nanoparticles were purchased from Sigma (Saint Quentin Fallavier, France) (catalog number #S5505). They were suspended in water as a 50 mg/mL stock solution and dispersed by sonication in a water sonication bath (BPAC-06VG). The suspension was then sterilized by pasteurization at 80 °C overnight. Prior to use, the suspension was homogenized by vortexing and bath sonication for 15 min.

The hydrodynamic size of the particles was determined after dilution in water or in a complete culture medium by dynamic light scattering (DLS) using a Wyatt Dynapro Nanostar instrument.

The morphology of nanoparticles was observed by TEM. Samples were absorbed to the clean side of a carbon film on mica and transferred to a 400-mesh copper grid. The images were taken at the magnifications of ×11K, ×13K, ×23K, and ×30K times with defocus values between 1.2 and 2.5 μm on an FEI Tecnai 12 LaB6 electron microscope (Hillsboro, OR, USA) at 120 kV accelerating voltage using a Gatan Orius 1000 CCD Camera (Pleasanton, CA, USA).

### 2.2. Cell Culture

The mouse macrophage cell line RAW 264.7 was obtained from the European Cell Culture Collection (Salisbury, UK). The cells were cultured in an RPMI1640 medium and 10% fetal bovine serum (FBS). For routine culture, cells were seeded on non-adherent flasks (i.e., suspension culture flasks from Greiner) at 200,000 cells/mL and harvested 48 h later at 1,000,000 cells/mL. Cell viability was measured by a dye exclusion assay, either with eosin (1 mg/mL) under the microscope or with propidium iodide (1 µg/mL) in a flow cytometry mode.

For treatment with pyrolytic silica, cells were seeded at 500,000 cells/mL and left for 24 h at 37 °C for cell adhesion. The medium volume was adjusted to keep a constant medium height for all the culture supports used (2 mL for 6-well plates and 15 mL for 75 cm^2^ flasks). The cells were grown to confluence for 48 h. The cells were then treated using the scheme described in Figure 1, as used before for silver nanoparticles [15]; in this study, however, we used 20 µg/mL pyrolytic silica.

In the control condition (center), cells were not exposed to silica. In the acute exposure condition (top row), cells were exposed to 20 µg/mL silica at day 3 post-seeding and sampled after 24 h of exposure. In the recovery exposure condition (bottom row), cells were exposed to 20 µg/mL silica at day 1 post-seeding, and then at day 2, the exposure medium was replaced by a fresh cell culture medium and cells were sampled at day 4.

### 2.3. Enzyme Assays

The cell extracts for enzyme assays were prepared as described previously [15]. The dehydrogenases or dehydrogenase-coupled activities were assayed at 500 nm using the phenazine methosulfate/iodonitrotetrazolium coupled assay [15]. Biliverdin reductase was assayed at 450 nm as described previously [17]. Pyridoxal kinase was assayed directly at 388 nm [18]. Pyruvate kinase was assayed by a decrease of nicotinamide adenine dinucleotide hydrate (NADH) at 340 nm in a lactate dehydrogenase-coupled assay [19]. Phosphoglycerate kinase was assayed by a decrease of NADH at 340 nm in a glyceraldehyde phosphate dehydrogenase (GAPDH)-coupled assay [20].

### 2.4. Phagocytosis and Particle Internalization Assay

The phagocytic activity was measured using fluorescent latex beads (1 µm diameter, green labelled, catalog number L4655 from Sigma) with the exclusion of the dead cells from the analysis, as described previously [15].

### 2.5. Mitochondrial Transmembrane Potential Measurement

The mitochondrial transmembrane potential was assessed by Rhodamine 123 uptake at low concentration (80 nM) to avoid quenching as described previously [15].

### 2.6. Lysosomal Function Evaluation

Ratiometric acridine orange fluorescence was used to investigate the lysosomal function. Cells were seeded into 6-well plates and exposed to the indicated dose of pyrolytic silica for twenty-four hours at 37 °C followed or not by a recovery period without nanoparticles. Acridine orange (Sigma A6529) was added to the cell culture (100 ng/mL), and the culture returned to the incubator for 30 min. Then, cells were harvested and washed with phosphate buffer saline-glucose (PBSG). The pellets were suspended with 250 μL of PBSG supplemented with Sytox Red (5 nM) and analyzed by flow cytometry. Acridine orange was excited at 475 nm, and the 526 and 650 nm emissions were recorded. The ratio between the two fluorescence intensities (red/green) was then calculated and used as an index of the lysosomal function [21].

### 2.7. NO Production and Cytokines Production

The cells were grown to confluence in 6-well plates and treated with silica, as described in Section 2.2. Then, half of the wells were treated with 100 ng/mL lipopolysaccharide (LPS) (from *Escherichia coli*, Sigma #L2880), and arginine monohydrochloride was added to all the wells (5 mM final concentration) to give a high concentration of substrate for the nitric oxide synthase. After 18 h of incubation, the nitrite concentration in the medium was assayed with the Griess reagent, as described previously [15].

For cytokine production, a commercial kit **(**BD Cytometric Bead Array, catalog number 552364 from BD Biosciences) was used. The supernatant of cells treated with silica was recovered and analyzed as indicated in the kit protocol.

### 2.8. F-Actin Staining

The experiments were performed essentially as previously described [15] using Phalloidin-Atto 550 (Sigma). The cells were cultured on coverslips placed in 6-well plates and exposed as described in Section 2.2. At the end of the exposure time, cells were washed and fixed in 4% paraformaldehyde for 30 min at room temperature. The cells were then processed as described previously [15].

### 2.9. Proteomics

The 2D gel-based proteomic experiments were carried out as previously described [15] on three independent biological replicates.

For protein identification, the MS/MS data were interpreted using a local Mascot server with the MASCOT 2.5.1 algorithm (Matrix Science, London, UK) against UniProtKB/SwissProt (version 2018_11,558,898 sequences). The research was carried out in all species. The protein identification parameters have been described previously [15]. Pathway analysis was performed using the DAVID tool [22].

## 3. Results

### 3.1. Nanoparticles Characterization and Determination of the Effective Doses

As shown in Figure 2A, the pyrolytic silica particles appeared as non-spherical aggregates of primary particles in the 5–10 nm size range. The hydrodynamic diameter of the aggregates, determined by dynamic light scattering (DLS), was close to 250 nm (mean value of 252 nm and polydispersity index of 0.25) in water and the zeta potential was measured at −11.27 ± 0.29 mV. This value did not appreciably change with time. In the complete culture medium (containing fetal calf serum), the starting value was similar to that observed in water (278 nm), but some aggregation took place, as indicated by a highly multimodal signal. Furthermore, the size of the aggregates increased over time, with an average value of 912 nm after 24 h of incubation in the complete cell medium (Figure 2B).

The effect of the pyrolytic silica particles on cell viability was then determined. The results, shown in Figure 2C, indicated a lethal dose 20% (LD20) of around 20 µg/mL, and this concentration was used for all further experiments.

### 3.2. Proteomic Studies

In order to gain further insights into the molecular responses of cells to the pyrolytic silica nanoparticles, we performed proteomic studies using two-dimensional electrophoresis, and annotated 2D gel images are shown in Appendix A. We first used the proteomic results globally, as described previously, by selecting a subset of proteins showing a variation (*p* < 0.25) in either the acute vs. control or the recovery vs. control comparisons [15]. This dataset was then analyzed by hierarchical clustering using the PAST statistical software suite [23]. The results, shown in Figure 3, indicate that the recovery and control conditions co-clustered, except for the acute exposure condition.

For a more detailed analysis of the proteomic results, we then selected proteins with significantly varied abundances (*p* ≤ 0.05) in at least one of the two comparisons (acute vs. control or recovery vs. control). The list of the selected proteins, together with their identification data, is given in Appendix A, and a selection of modulated proteins is displayed in Table 1.

Consistent with the results obtained at the global analysis stage, only a minority of proteins (highlighted in bold in Appendix A) showed an abundance change at the end of the recovery period greater than the change observed at the end of the exposure period. In order to get an integrated view of the results, a pathway analysis was performed using the DAVID tool [22]. A simplified view of the results is given in Table 2, and the detailed results are presented in Appendix A. Several pathways were altered (e.g., cell adhesion actin cytoskeleton, carbon metabolism, mitochondrion, and proteasome), which led to validation studies, as described below.

### 3.3. Validation Studies

In addition to the data obtained from the pathway analysis, we used previous knowledge obtained on other nanoparticles, such as colloidal amorphous silica [11] or silver nanoparticles [15] to select the biological parameters to be investigated in the validation studies.

#### 3.3.1. Enzyme Activities

Carbon metabolism—especially glycolysis—was one of the pathways highlighted in the pathway analysis, with several glycolytic enzymes involved such as lactate dehydrogenase (P06151), glyceraldehyde 3-phosphate dehydrogenase (P16858), phosphoglycerate kinase (P09411), or pyruvate kinase (P52480). In order to validate the proteomic results, the activity of these enzymes was measured, and the results are displayed in Table 3. The activity measurements were consistent with those obtained through proteomics. For the glycolytic enzymes, a major decrease in activity was observed immediately after exposure to pyrolytic silica, with a restoration of normal or close to normal activities at the end of the recovery period. For the two non-glycolytic enzymes tested (biliverdin reductase and pyridoxal kinase), there was a good correlation between the proteomic and activity data at the end of the exposure period. The concordance was less pronounced at the end of the recovery period.

#### 3.3.2. Cytoskeleton and Phagocytosis

Several proteins associated with the actin cytoskeleton, such as cofilin (P18760) and cap1 (P40124), emerged from the proteomic screen, as did capping or debranching proteins (P47754, P24452, and Q9CQI3). Consistent with these observations, “actin cytoskeleton” was one of the terms appearing in the pathway analysis (Appendix A). This led us to study whether the actin cytoskeleton was altered in macrophages upon treatment with silica. The results, displayed in Figure 4A–C, showed that pyrolytic silica induced a decrease in the surface of the cytoplasmic projections (lamellipodia and filipodia) observed at the surface of the macrophages, as previously observed with colloidal silica [11]. This effect was persistent at the end of the 72 h recovery period. As the actin cytoskeleton is also involved in phagocytosis, we also tested this macrophage function. The results, displayed in Figure 4D, showed no significant effect of the exposure of cells to pyrolytic silica, either immediately after exposure or at the end of the recovery period.

#### 3.3.3. Mitochondrial Potential

A few mitochondrial proteins were also detected by the proteomic screen, among which a chaperone (O35501), a stress response protein (P20108), a Krebs cycle protein (Q9D2G2), prohibitin (P67778), TMEM11 (Q8BK08), a mitochondrial deamidase (Q8VDK1), a subunit of respiratory complex I (Q9DCT2), and an ATP synthase subunit (P56480). This led us to test whether the mitochondrial transmembrane potential was altered upon exposure to pyrolytic silica. The results showed no significant effect, either in the proportion of rhodamine-positive cells (>90% in all cases) or in the intensity of the rhodamine signal (range of 942–1187 fluorescence units), which is an indirect index of the mitochondrial transmembrane potential.

#### 3.3.4. Lysosomal Integrity

A few proteins implicated in endosomal/lysosomal function, such as AP3-mu1 (Q9JKC8), cathepsin S (O70370), CHM2A (Q9DB34), hook 3 (Q8BUK6) serpin b6 (Q60854), sorting nexin 6 (Q6P8X1), VATA (P50516), and VPS29 (Q9QZ88) also emerged from the proteomic screen. This prompted us to test the lysosomal integrity via the acridine orange ratiometric method [21]. The red/green fluorescence intensity ratios were as follows:Control cells: 0.9884 ± 0.0116.Cells acutely exposed to 20 µg/mL pyrolytic silica for 24 h and analyzed just after exposure: 0.9888 ± 0.0038.Cells acutely exposed to 20 µg/mL pyrolytic silica for 24 h and analyzed after a 72 h recovery period: 1.0001 ± 0.0044.These data showed no significant perturbation in the lysosomal integrity under the conditions tested.

#### 3.3.5. Inflammatory Responses

As silica is known to exert a proinflammatory effect on macrophages, the persistence of which differs between amorphous and crystalline silica, we tested this parameter in two different schemes. With cells unstimulated by lipopolysaccharide, we tested the intrinsic proinflammatory effects of the pyrolytic silica. The results, displayed in Figure 5A, showed no significant change in the NO production upon treatment with pyrolytic silica or during the recovery phase. Regarding cytokine secretion, the persistence of the effect differed to some extent for the two measured cytokines. Tumor necrosis factor alpha (TNF) levels showed a trend to return to basal levels, but the levels were still significantly higher at the end of the recovery period than in unexposed cells (Figure 5D, black bars). The production of interleukin 6 (IL-6) did not decrease (and even increased slightly) during the recovery period compared to acute exposure (Figure 5B). Thus, the results obtained on both TNF and IL-6 showed a persistence of the intrinsic pro-inflammatory effect of pyrolytic silica.

With cells stimulated with lipopolysaccharide, we tested the interference that exposure to silica may bring to macrophage responses to bacteria. The results, displayed in Figure 5C,D (grey bars), showed an increase in the production of IL-6 immediately after exposure, and this was lost at the end of the recovery period. No significant effects were observed on nitric oxide or on TNF production.

## 4. Discussion

Though both are classified as amorphous silica and used at levels of millions of tons per year worldwide, pyrolytic silica and silica produced by a wet route seem to have different effects on cells, at least at the inflammatory level [7,8]. Using a proteomic screen and comparing the present data obtained on pyrolytic silica with the ones obtained on the same cell line but with a colloidal silica [11] allowed us to widen the comparison of the relative effects of the two silica types on macrophages. Though the modulated proteins can be different between the two types of amorphous silica (Appendix A), some pathways are clearly shared for both types of amorphous silica, as shown by the comparison of the pyrolytic (Appendix A) and colloidal (Appendix A) pathway analyses. For example, a clear induction of several proteasome subunits was shown in both cases. Different proteins involved in the control of the actin cytoskeleton were also modulated by the colloidal and pyrolytic silica, with the same outcome of the reduction of the cytoplasmic projections. This reduction was, however, not linked to particle internalization itself, as it was not observed for zirconium oxide nanoparticles (e.g., [24]), thus showing the good correlation between the results of the proteomic screen and the phenotypic result.

Going into more detail, for most of the proteins that showed a significant change in their abundance upon treatment with both types of silica, the same type of change was observed for pyrolytic (this study) and colloidal silica [11]. This consistency held true for bpnt1 (Q9Z0S1), bvra (Q9CY64), fkbp4 (P30416), fpps (Q920E5), pddc1 (Q8BFQ8), phb (P67778), spre (Q64105), ssrd (Q62186), and twf2 (Q9Z0P5), i.e., proteins belonging to very different pathways. This documented the fact that part of the cellular response is common for both types of amorphous silica. However, the proteomic screen also revealed many differences in the cellular response to the two types of silica (see Appendix A), e.g., a major decrease in glycolytic enzymes observed for pyrolytic silica but not for colloidal silica.

Going on with cellular energetics, several abundance changes were noticed by proteomics for mitochondrial proteins at the end of the exposure period, while the mitochondrial function appeared normal. In this context, it should be noted that the abundance increased, especially for proteins involved in mitochondrial maintenance such as prohibitin [25] and TMEM11 [26], suggesting a successful adaptation of the cells to the stress induced by the uptake of pyrolytic silica.

This cellular response to stress was also documented at the proteomic level by increases in the abundance of the deglycase DJ-1 (Q99LX0), formylglutathione hydrolase esd (Q9R0P3), and protein pdcd6 (P12815), which is involved in membrane repair [27].

Phenotypically, the results described here with one type of pyrolytic silica were consistent with those described with the reference pyrolytic silica material NM203 and macrophages [8]. In particular, the inflammatory responses (the production of IL-6 and TNF) were similar in both cases, showing the consistency of the macrophages’ responses toward different pyrolytic silica.

However, most results obtained in vitro on amorphous silica and present in the literature have used an acute exposure scheme where the cellular response was analyzed immediately after exposure. Such a scheme does not allow for the investigation of the persistence of the cellular effects. Because amorphous silica and especially pyrolytic silica have shown persistent effects in vivo [3,28], it was interesting to set up an in vitro system to help to investigate the effect persistence. We used a system where the exposure was followed by a recovery period, during which the persistence of the effects could be investigated. As macrophages are known to internalize various types of particulate materials, including amorphous silica (e.g., in [16]), and keep insoluble materials internalized for extended periods of time [29,30], this system investigated the longer-term responses of cells to the internalized nanomaterials.

This is why we previously used this system to investigate the persistent effects of silver nanomaterials, e.g., nanowires [12] and nanoparticles [15]. When used on pyrolytic silica, this system demonstrated a persistence of the inflammatory response (IL-6 and TNF) in line with the prolonged inflammation described in vivo [3,28]. Such a persistence was not observed with a colloidal silica [12], which was used as a control.

Going into more detail about this system, we selected macrophages as the target cell type because they are the scavenger cell type for particulate materials, are found in numerous locations in the body, and are key players in inflammatory reactions [31]. Mouse macrophage cell lines show excellent responses to many stimuli and can be used in various exposure scenarii such as chronic exposures [32]. For such long-term experiments and for medium-term experiments such as the recovery scheme used here, the cells must be cultured in a complete culture medium containing serum because they will not survive for such extended period of times in serum-free media. Because the formation of a protein corona is known to alter the toxicity of amorphous silica [33,34], the relevance of our system for toxicological evaluation purposes can be questioned. In this frame, it should be kept in mind that all biological fluids, including pulmonary surfactant [35,36,37], contain rather high concentrations and a large variety of proteins. Thus, it seems to us that the culture conditions used should be a reasonable model of the conditions occurring in vivo, both in terms of nanomaterial aggregation and cellular physiology. Indeed, our in vitro results (the intrinsic induction of IL-6 and TNF, low effects in the case of silica-LPS co-exposure, no effect on phagocytic capacity) paralleled-well to those obtained in vivo in a model of pneumonia [38]. This shows that our system is not plagued by gross artifacts. Moreover, it should be kept in mind that exposure to amorphous silica does not only occur by the pulmonary route, as amorphous silica is also used as a food additive in cosmetics and in toothpastes. For such non-pulmonary routes where macrophages are located below the body barriers, there is no question about the presence of high concentrations of proteins when the nanomaterials come in contact with the macrophages.

When focusing on the recovery phase, the proteomic study showed that many cellular functions had returned to normal levels at the end of the recovery period. In this respect, the recovery was much more obvious for pyrolytic silica than for silver nanoparticles [15]. It was then of interest to study the few proteins for which a stronger change of abundance was observed at the end of the recovery period rather than immediately after exposure to pyrolytic silica. Among these few proteins, an early trend emerged in proteins implicated in the endosomal/lysosomal pathway, mostly proteins implicated in vesicular traffic such as CHM2A (Q9DB34) [39], hook 3 (Q8BUK6) [40] sorting nexin 6 (Q6P8X1) [41], and VPS29 (Q9QZ88) [42]. The hook 3 protein also binds the scavenger receptor [43], which is known to be implicated in silica internalization [44,45]. In addition, Serpin B-6 (Q60854) was also increased during the recovery phase. This protein is known to inhibit lysosomal proteases [46] and may play a role against cytosolic damage induced by the liberation of lysosomal proteases, thus prompting us to investigate lysosomal integrity. While lysosomal rupture has been described in the case of crystalline silica-induced apoptosis [47], i.e., at high doses, we did not detect any massive lysosomal rupture under our conditions, which was in line with what has been previously described with amorphous silica on the same RAW264.7 cells at non-lethal concentrations [48].

Additionally noteworthy was an increase in two detoxifying proteins, aldo-keto-reductases 1a1 (Q9JII6) and 1b1 (P45376). These proteins reduce a wide range of toxic electrophiles [49,50], including acrolein [51], glyceraldehyde [52], and methylglyoxal [50,53]. Thus, the induction of these detoxifying proteins may be linked with the observed impairment of glycolysis during acute exposure. Glycolysis inhibition increases the concentration of toxic aldehydes, as exemplified by zinc oxide nanoparticles [24]. Thus, we may observe a sequential process in which the glycolysis is impaired first, thus liberating aldehydes that must be destroyed during the recovery phase.

Overall, the fact that several biological parameters (e.g., phagocytic ability, mitochondrial potential, and lysosomal integrity) were not altered during the exposure to pyrolytic silica at a non-lethal dose suggested that the changes observed through the proteomic screen were successful adaptive changes.

## 5. Conclusions

In conclusion, the simple exposure/recovery in vitro system described here proved to be able to replicate a part of the inflammation persistence phenomena observed during in vivo studies [3,28] and to differentiate between the inflammatory effects of colloidal silica [12] and pyrolytic silica (this study). Coupling with proteomics allowed for a broader understanding of the molecular phenomena at play in the cells during both the exposure and recovery periods.

Such features are promising in the development of simple in vitro assays that may be used as first screens to limit the use of laboratory animals following the 3R principles that are now developing in some areas of the world, e.g., in the EU (http://eur-lex.europa.eu/legal-content/EN/TXT/PDF/?uri=CELEX:32010L0063&from=EN%7CDirective2010/63/EU of the European Parliament and of the Council on the protection of animals used for scientific purposes).

## Figures and Tables

**Figure 1 nanomaterials-10-01939-f001:**
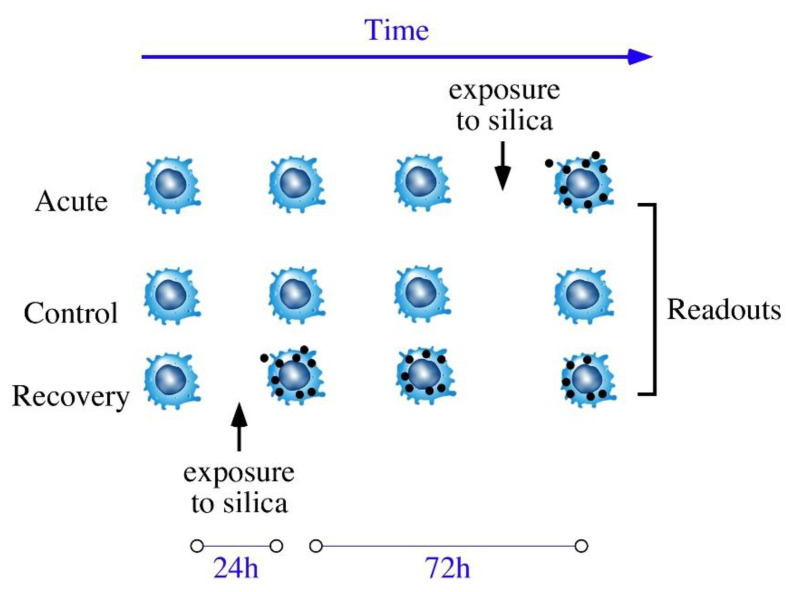
Schematic representation of the exposure schemes used in the study.

**Figure 2 nanomaterials-10-01939-f002:**
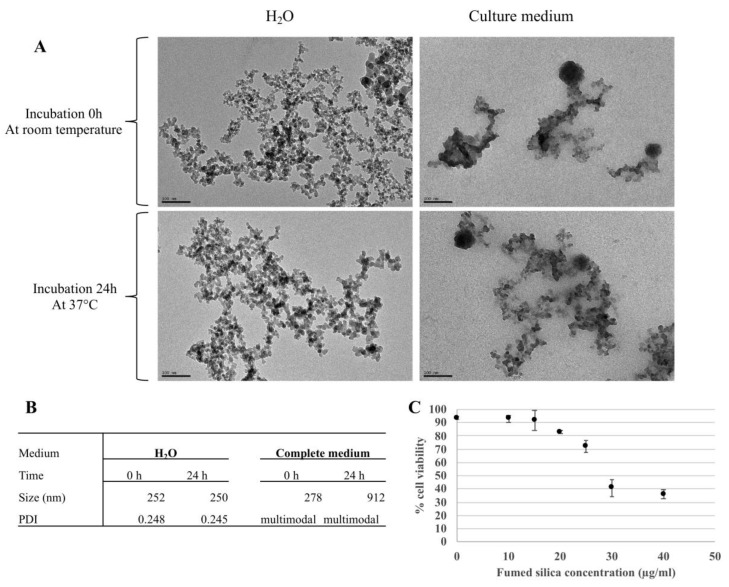
Pyrolytic silica characterization (by transmission electron microscope and dynamic light scattering) and cell viability with flow cytometer. (**A**) Observation by TEM. Nanoparticles were incubated in H_2_O (left) or culture medium (right) for either 0 h of incubation at room temperature (top) or 24 h of incubation at 37 °C and 5% CO_2_ (bottom). Scale bar = 100 nm. (**B**) Table showing measures by DLS of the hydrodynamic diameter of nanoparticles in water and a culture medium at 0 and after 24 h of incubation. PDI: polydispersity percentage. (**C**) Cell viability of RAW264.7 exposed to fumed silica for 24 h using propidium iodide (1 μg/mL).

**Figure 3 nanomaterials-10-01939-f003:**
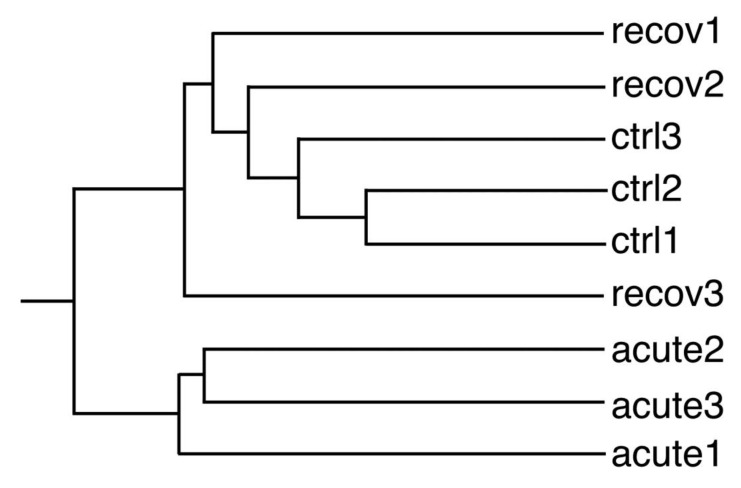
Global analysis of the proteomic experiment by hierarchical clustering. Quantitative proteomic data (i.e., spot intensities in 2D gel-based proteomics) were used to determine the similarities between biological samples. The PAST software package was used for calculations using the Gower similarity index and a paired group algorithm. This classification tree should be read from left to right. The earlier the bifurcation point, the more dissimilar the groups are. ctrl (1–3): unexposed cells; acute (1–3): cells exposed to pyrolytic silica for 24 h and collected just after exposure; recov (1–3): cells exposed to pyrolytic silica for 24 h and left to recover without silica for 72 h. Thus, this tree shows that proteome-wise, the cells just at the end of the exposure time were the most different from the other two conditions, while the three replicates for the acute condition were quite consistent among themselves. The three replicates for control cells were also consistent, while the replicates for the recovery condition were less consistent, indicating variability detected at the proteome scale.

**Figure 4 nanomaterials-10-01939-f004:**
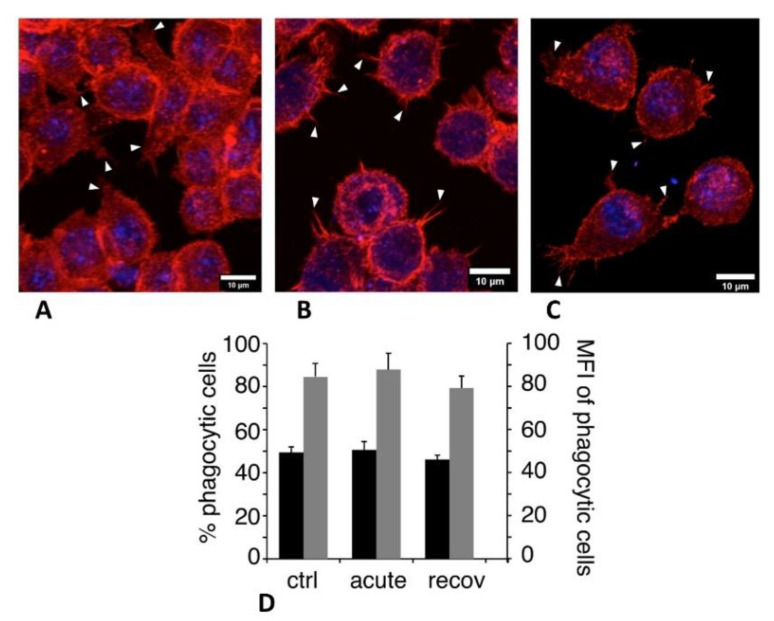
Actin cytoskeleton and phagocytosis. In panels (**A**–**C**), the actin cytoskeleton was visualized with fluorescent Atto 550 phalloidin (500 ng/mL) using LSM880 confocal microscopy. A Z-project reconstruction is shown for each condition. Examples of cytoplasmic projections (lamellipodia and filipodia) are indicated by white arrowheads. (**A**) Unexposed cells. (**B**) Cells exposed to pyrolytic silica for 24 h and analyzed just after exposure. (**C**) Cells exposed to pyrolytic silica for 24 h and left to recover without silica for 72 h. In panel (**D**), the phagocytic capacity was assessed by fluorescent latex beads internalization. Black bars: proportion of positive cells in the viable cell population; grey bars: mean cellular fluorescence of positive cells; ctrl: unexposed cells; acute: cells exposed to pyrolytic silica for 24 h and collected just after exposure; and recov: cells exposed to pyrolytic silica for 24 h and left to recover without silica for 72 h.

**Figure 5 nanomaterials-10-01939-f005:**
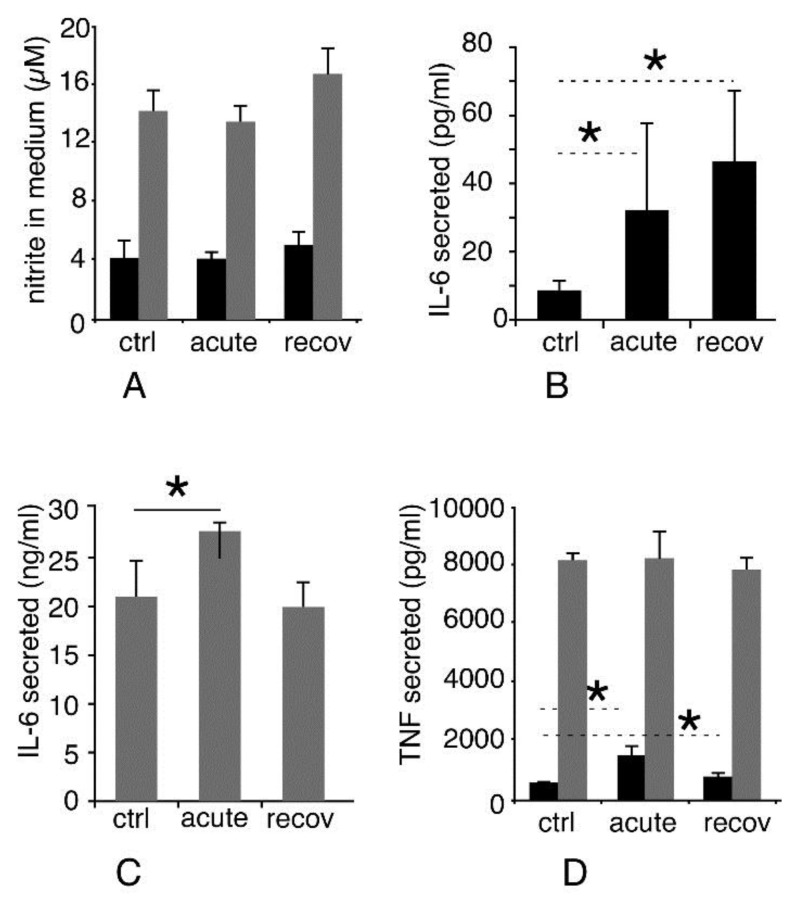
Inflammatory responses. The secretion of NO, interleukin-6, and TNF alpha was measured in response to pyrolytic silica, either alone or in combination with LPS. Ctrl: unexposed cells; acute: cells exposed to pyrolytic silica for 24 h and collected just after exposure; and recov: cells exposed to pyrolytic silica for 24 h and left to recover without silica for 72 h. (**A**) Production of nitric oxide. Black bars: production after treatment for 24 h with pyrolytic; grey bars: production after treatment for 24 h with pyrolytic silica and stimulation with LPS for the last 18 h of culture. (**B**) Production of interleukin-6 in the absence of LPS stimulation. (**C**) Production of interleukin-6 in the presence of LPS stimulation. (**D**) Production of TNF. Black bars: production after treatment for 24 h with pyrolytic silica; grey bars: production after treatment for 24 h with pyrolytic and stimulation with LPS for the last 18 h of culture. Symbols indicate the statistical significance (Mann–Whitney U-test): * *p* < 0.05.

**Table 1 nanomaterials-10-01939-t001:** Selection of differentially-expressed proteins identified in the proteomic screen.

Code	Name	Accession	Ratio/*t*-Test Acute/Control	Ratio/*t*-Test Recov./Control
***akr1a1***	***Alcohol dehydrogenase [NADP(+)]***	***Q9JII6***	***1.23/0.39***	***0.75/0.07***
**akr1b1**	**Aldose reductase**	**P45376**	**0.91/0.34**	**0.83/0.04**
ap3m1	AP-3 complex subunit mu-1	Q9JKC8	2.23/0.01	0.79/0.64
atpb	ATP synthase subunit beta, mitochondrial	P56480	1.55/0.01	1.22/0.27
bpnt1	3′(2′),5′-bisphosphate nucleotidase 1	Q9Z0S1	2.09/0.02	1.28/0.37
bvra ac	Biliverdin reductase A	Q9CY64	0.41/0.04	1.01/0.98
cap1	Adenylyl cyclase-associated protein 1	P40124	0.69/0.03	1.01/0.96
capg	Macrophage-capping protein	P24452	0.82/0.04	0.90/0.31
**casp3**	**Caspase-3**	**P70677**	**0.99/0.99**	**0.68/0.04**
catS	Cathepsin S	O70370	0.71/0.02	0.91/0.33
caza2	F-actin-capping protein subunit alpha-2	P47754	1.31/0.02	0.96/0.56
**chm2a**	**Charged multivesicular body protein 2a**	**Q9DB34**	**0.95/0.40**	**0.65/0.01**
*cof*	*Cofilin-1*	*P18760*	*0.52/0.06*	*0.95/0.88*
*dj1*	*Protein deglycase DJ-1*	*Q99LX0*	*1.65/0.09*	*0.93/0.77*
esd	S-formylglutathione hydrolase	Q9R0P3	1.26/0.03	0.86/0.10
*fkbp4*	*Peptidyl-prolyl cis-trans isomerase FKBP4*	*P30416*	*1.56/0.11*	*0.91/0.72*
fpps	Farnesyl pyrophosphate synthase	Q920E5	1.40/0.04	1.25/0.35
gapdh	Glyceraldehyde-3-phosphate dehydrogenase	P16858	0.56/0.01	1.03/0.85
gmfb	Glia maturation factor beta	Q9CQI3	0.75/0.03	1.22/0.29
grp75	Stress-70 protein, mitochondrial	O35501	1.94/0.007	1.02/0.89
**hook3**	**Protein Hook homolog 3**	**Q8BUK6**	**1.07/0.65**	**0.45/0.005**
kpym	Pyruvate kinase PKM	P52480	0.66/0.048	0.98/0.88
ldha	L-lactate dehydrogenase A chain	P06151	0.60/0.00675	0.91/0.64
ndus3	NADH dehydrogenase [ubiquinone] iron-sulfur protein 3, mitochondria	Q9DCT2	1.76/0.01	1.06/0.71
nit1	Nitrilase homolog 1	Q8VDK1	0.53/0.01	0.82/0.10
odo2	Dihydrolipoyllysine-residue succinyltransferase component of 2-oxoglutarate dehydrogenase complex, mitochondrial	Q9D2G2	1.33/0.02	0.94/0.80
pdcd6	Programmed cell death protein 6	P12815	1.32/0.04	1.06/0.53
**pddc1**	**Parkinson disease 7 domain-containing protein 1**	**Q8BFQ8**	**0.70/0.20**	**0.77/0.01**
pgk1	Phosphoglycerate kinase 1	P09411	0.70/0.03	0.95/0.50
phb	Prohibitin	P67778	1.27/0.0018	0.98/0.78
prx3	Thioredoxin-dependent peroxide reductase, mitochondrial	P20108	1.83/0.02	0.82/0.37
psa1	Proteasome subunit alpha type-1	Q9R1P4	1.35/0.03	0.86/0.06
psb10	Proteasome subunit beta type-10	O35955	1.45/0.03	0.64/0.06
psb4	Proteasome subunit beta type-4	P99026	2.90/0.04	1.02/0.93
psmd2	26S proteasome non-ATPase regulatory subunit 2	Q8VDM4	1.69/0.04	1.05/0.79
psmd14	26S proteasome non-ATPase regulatory subunit 14	O35593	1.51/0.05	0.91/0.61
**snx6**	**Sorting nexin-6**	**Q6P8 × 1**	**0.97/0.53**	**0.88/0.007**
**spb6**	**Serpin B6**	**Q60854**	**0.83/0.11**	**1.23/0.02**
spre	Sepiapterin reductase	Q64105	0.76/0.03	1.21/0.39
tmem11	Transmembrane protein 11, mitochondrial	Q8BK08	1.50/0.02	1.21/0.55
twf2	Twinfilin-2	Q9Z0P5	0.68/0.03	0.92/0.65
vata	V-type proton ATPase catalytic subunit A	P50516	1.66/0.01	1.01/0.95
**vps29**	**Vacuolar protein sorting-associated protein 29**	**Q9QZ88**	**0.99/0.99**	**0.67/0.03**

Bold: proteins with a change greater in amplitude at the end of the recovery period compared to the change observed immediately after exposure. Italics: proteins showing a significant change by the Mann–Whitney U test but not by the Student *t* test.

**Table 2 nanomaterials-10-01939-t002:** Selection of modulated pathways highlighted by the DAVID functional annotation chart tool.

Term	Count ^1^	*p*-Value ^2^	FDR in % ^3^
GO:0005925~focal adhesion	7	0.0049	0.068
GO:0098609~cell–cell adhesion	5	0.0086	0.96
GO:0007005~mitochondrion organization	5	3.94 × 10^−4^	0.22
GO:0005739~mitochondrion	24	1.59 × 10^−7^	5.775 × 10^−6^
Proteasome	5	4.58 × 10^−5^	0.0016
Hydrolase	18	4.97 × 10^−5^	0.0016
Ubl conjugation	17	6.13 × 10^−5^	0.0016
GO:0000502~proteasome complex	5	1.44 × 10^−4^	0.0033
mmu01200: Carbon metabolism	7	1.64 × 10^−4^	0.011
GO:0006006~glucose metabolic process	4	0.0030	0.43
GO:0005975~carbohydrate metabolic process	6	0.0018	0.39
Glycolysis	3	0.0068	0.083
GO:0006096~glycolytic process	3	0.010	0.96
NAD	6	4.77 × 10^−4^	0.0089
Oxidoreductase	9	0.0018	0.029
IPR002108: Actin-binding, cofilin/tropomyosin type	3	6.57 × 10^−4^	0.082
Actin-binding	6	0.0019	0.029
GO:0051016~barbed-end actin filament capping	3	0.0021	0.39
GO:0003779~actin binding	7	0.0040	0.27
GO:0031982~vesicle	5	0.0043	0.065
GO:0005765~lysosomal membrane	5	0.015	0.18

^1^ Count: number of proteins in the total list of modulated proteins assigned to the selected pathway. ^2^
*p*-Value: probability for the selected pathway to be selected by random. ^3^ FDR (false discovery rate): *p*-Value corrected for multiple testing.

**Table 3 nanomaterials-10-01939-t003:** Enzyme activities.

Enzyme	Control	Acute	Recov
BVR	1.86 ± 0.62	0.58 ± 0.41 *	1.08 ± 0.26
GAPDH	19.83 ± 3.33	1.50 ± 1.00 *	19.50 ± 1.73
LDH	119.50 ± 16.26	23.17 ± 14.74 *	134.17 ± 10.27
PDXK	9.33 ± 2.91	15.33 ± 3.05	16.67 ± 5.29
PGK	3547 ± 546	2242 ± 1059	2950 ± 597
PKM	4219 ± 607	983 ± 581 *	4998 ± 195

All the activities are expressed in nmole substrate converted/min/mg total protein. Statistical significance of the results in the Student *t* test: * *p* < 0.05. Abbreviations: BVR: biliverdin reductase; GAPDH: glyceraldehyde phosphate dehydrogenase; LDH: lactate dehydrogenase; PDXK: pyridoxal kinase; PGK: phosphoglycerate kinase; PKM: pyruvate kinase.

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
