# Peer review of "How Reversible Are the Effects of Fumed Silica on Macrophages? A Proteomics-Informed View"

_nanomaterials, 2020, doi:10.3390/nano10101939_

Round 1

Reviewer 1 Report

The manuscript titled 'How reversible are the effects of fumed silica on 3 macrophages? A proteomics-informed view' is an interesting and relevant study which sheds light on the cellular system to revert back to normal biological state subsequent to treatment by pyrolytic silica and recovery period.

Here are a few suggestions to be included:

  1. Could you expand on 'unexpected results' in line 53 and 54 - it leaves the reader guessing until they read paper 5 and 7.
  2. Revise line 159-160
  3. F-actin staining could have been measured with flow cytometry for quantitative set of data - was this considered? The changes in the number of spike does not seem clear.
  4. line 275 - copper? 
  5. line 281 -confusion with using ± 72 hours - the recovery period was 72 hours ?
  6. lines 301 - 304 - change sentence as it is not clear
  7. legends need to be consistent and descriptive
  8. It would be useful to have a summary table for similarities/ differences between pyrolytic silica vs. colloidal silica. Some proteins are mentioned in the discussion which could comprise the table with various proteins in rows and colloidal and pyrolytic as columns. This would add some clarity and focus on the importance of the study. 

Author Response

First and foremost, we would like to thank warmly Reviewer 1 for the very useful report that he/she sent to us. We would like to take some space to detail our responses to the specific concerns risen by Reviewer 1. Please note that in the revised version, the changes made appear in blue text color

comment

The manuscript titled 'How reversible are the effects of fumed silica on 3 macrophages? A proteomics-informed view' is an interesting and relevant study which sheds light on the cellular system to revert back to normal biological state subsequent to treatment by pyrolytic silica and recovery period.

Here are a few suggestions to be included:

  1. Could you expand on 'unexpected results' in line 53 and 54 - it leaves the reader guessing until they read paper 5 and 7.

Reply

The major results of these two papers are now described in lines 58-59 and 62-63 (in blue)

Comment

  1. Revise line 159-160

Reply

This was in the legend of Figure 1. The legend of now figure 2 has been modified

Comment

  1. F-actin staining could have been measured with flow cytometry for quantitative set of data - was this considered? The changes in the number of spike does not seem clear.

Reply

In fact the total amount of actin does not change, and proeomics detects indeed no change in it. It is rather the organization that changes. Furthermore, macrophages adhere strongly to surfaces, and detaching them from the surface was likely to lead to cytoskeletal reorganization. We provide a new confocal microscopy figure, made on the same coverslips but with a different microscopy procedure, to better show our point. Furthermore, we have tried to be more precise in the description made (lines 261-262, in blue)

Comment

  1. line 275 - copper? 

Reply

That was a mixup with another paper that we are writing on macrophages and copper. Thank you for pointing out the mistake, which we corrected

Comment

line 281 -confusion with using ± 72 hours - the recovery period was 72 hours ?

Reply

We have changed the legend of now figure 5 for better clarity

Comment

lines 301 - 304 - change sentence as it is not clear

Reply

We have changed the sentence (now lines 352-355, in blue)

Comment

legends need to be consistent and descriptive

Reply

We have checked and changed the legends for better clarity

Comment

It would be useful to have a summary table for similarities/ differences between pyrolytic silica vs. colloidal silica. Some proteins are mentioned in the discussion which could comprise the table with various proteins in rows and colloidal and pyrolytic as columns. This would add some clarity and focus on the importance of the study

Reply

Thank you very much for this suggestion. To this purpose, we have added Tables S3 and S4, and changed the text of the discussion according to the results shown in these tables (lines 352-355, in blue)

Reviewer 2 Report

In the current article Torres et al have investigated the acute a chronic cellular response after exposure to synthetic amorphous nanoparticles. The authors have evaluated the macrophage biological response after 24hours of continuous exposure and a recovery phase of 72h post-exposure in a nanoparticle free media. The foundation of the study is a 2D gel proteomics screen where the authors have identified protein change clusters and then validate pathway changes in the cell line making the study of high interest. 

The reviewer feels that the article is of great interest but has the following questions:

  1. The reviewer feels that the introduction requires more context when describing the possible real hazard (for example to workers exposed to fume silica) and the reason why amorphous vs crystalline silica may lead to a different biological response. Additionally, pyrolytic silica is suddenly mentioned in the introduction without any additional background. The reviewer agrees that the exposure to pyrolytic silica is a matter of concern and must be investigated, but the experimental design is not straightforward and needs an additional background. In particular, a typical experiment would require the nanoparticle exposure to a pulmonary cell line to mimic the aerosol exposure, but working with macrophages is also important in case of the nanoparticle translocation into the bloodstream as it would lead to an immunogenic response. Therefore, the reviewer feels that this link would help the reader in understanding the experimental design.
  2. The researcher will also investigate the biological response after a recovery period. While this study is not often studied, it would help the reader the presence of a schematic figure or additional information at the end of the introduction (page 2 line 64).
  3. In the materials and methods section, the authors refer to the published articles but the essential experimental information should be present in the article.
  4. In section 2.1 the authors refer to fumed silica nanoparticles. For clarity, it would be less confusing whether the particles are either called pyrolytic silica or fumed silica across the manuscript.
  5. In section 2.2 line 92, the authors refer to a scheme that was published in a cited reference. The reviewer suggests either to include the scheme or describe it in words.
  6. The physicochemical data shows strong particle aggregation and colloidal instability over time. Additional aggregation occurs after nanoparticle exposure to culture media, most likely due to high salt concentration and by the presence of proteins.  Additionally, biomolecules from the environment will bind to the nanoparticle surface forming a biological corona and it will also have an effect in the biological response. The reviewer feels that the authors should discuss more these aspects in the discussion. This is the major drawback of the study and the reviewer feels that this limit should be openly mentioned in the manuscript.
  7. The polydispersity index of the DLS data is conventionally reported as a value from 0 to 1 rather than in percentage.
  8. Figure 2 is not well described in the manuscript text. The reviewer suggests either to provide a more detailed explanation of the figure highlighting the main findings or to move it to the SI.

Minor comments

Figure 1. each figure panel should be labelled with a letter and should not be grouped  

The reviewer found inconsistency with the font and size used in the figures (either on the axes or when labelling the panel).

Figure 1b is a table and should be used as table 1 rather than a figure

Please go through the manuscript for typos and use italic (eg in vitro) and subscript when appropriate (eg C02 should be CO2)

Author Response

First and foremost, we would like to thank warmly Reviewer 2 for the very useful report that he/she sent to us. We would like to take some space to detail our responses to the specific concerns risen by Reviewer 2. Please note that in the revised version, the changes made appear in blue text color

Comment

The reviewer feels that the introduction requires more context when describing the possible real hazard (for example to workers exposed to fume silica) and the reason why amorphous vs crystalline silica may lead to a different biological response. Additionally, pyrolytic silica is suddenly mentioned in the introduction without any additional background. The reviewer agrees that the exposure to pyrolytic silica is a matter of concern and must be investigated, but the experimental design is not straightforward and needs an additional background. In particular, a typical experiment would require the nanoparticle exposure to a pulmonary cell line to mimic the aerosol exposure, but working with macrophages is also important in case of the nanoparticle translocation into the bloodstream as it would lead to an immunogenic response. Therefore, the reviewer feels that this link would help the reader in understanding the experimental design.

Reply

We have expanded these aspects in the introduction (now lines 38-47, in blue)

Comment(s)

The researcher will also investigate the biological response after a recovery period. While this study is not often studied, it would help the reader the presence of a schematic figure or additional information at the end of the introduction (page 2 line 64).

Plus

In section 2.2 line 92, the authors refer to a scheme that was published in a cited reference. The reviewer suggests either to include the scheme or describe it in words.

Reply

We have added now Figure 1 in the material and methods section to address these comments

Comment

In the materials and methods section, the authors refer to the published articles but the essential experimental information should be present in the article.

Reply

In this matter, we are constrained by the rules imposed by the publisher on self-plagiarism, so that this matter is beyond our control. We have however taken care of citing only open access papers, so that the detailed methods are available to any interested person.

Comment

In section 2.1 the authors refer to fumed silica nanoparticles. For clarity, it would be less confusing whether the particles are either called pyrolytic silica or fumed silica across the manuscript.

Reply

Fumed silica has been replaced by pyrolytic silica everywhere in the manuscript

Comment

The physicochemical data shows strong particle aggregation and colloidal instability over time. Additional aggregation occurs after nanoparticle exposure to culture media, most likely due to high salt concentration and by the presence of proteins.  Additionally, biomolecules from the environment will bind to the nanoparticle surface forming a biological corona and it will also have an effect in the biological response. The reviewer feels that the authors should discuss more these aspects in the discussion. This is the major drawback of the study and the reviewer feels that this limit should be openly mentioned in the manuscript.

Reply

This is a very interesting question, that turns in how to translate at best in vitro the in vivo situation. We have addressed this question in the discussion, lines 401-420 in blue

Comment

The polydispersity index of the DLS data is conventionally reported as a value from 0 to 1 rather than in percentage.

Plus

Figure 1b is a table and should be used as table 1 rather than a figure

Reply

Polydispersity index changed in now Figure 2. We believe it is easier for the reader to have all data simultaneously at view in one Figure instead of split between a figure and a table. This is why we insist in keeping this format

Comment

Figure 2 is not well described in the manuscript text. The reviewer suggests either to provide a more detailed explanation of the figure highlighting the main findings or to move it to the SI.

Reply

We have expanded the description of now Figure 3, especially in its legend

Comment

Please go through the manuscript for typos and use italic (eg in vitro) and subscript when appropriate (eg C02 should be CO2)

Reply

We have combed the manuscript and made the corrections

Reviewer 3 Report

The manuscript by Torres et al. investigates the effects of a persistent exposure to pyrolytic silica, adopting a recovery approach in which silica NP are “removed” from cultured cells. The Authors have assessed the effects of pyrolytic silica NP combining omic and targeted methods using the murine macrophage cell line Raw 264.7. The subject of the study, as well as the analytical approaches adopted to achieve the objectives of the work, are potentially interesting and they could provide novel findings to understand the toxicological properties of amorphous silica NP.

However, the manuscript has several methodological and conceptual flaws that should be faced by the Authors.

The following major points should be properly addressed:

- When the Authors talk about “recovery from silica NP”, they simply replace the NP-additioned media with NP-free media. However, they do not consider the sedimentation of the NP on the cells. Because NP form aggregates, it is impossible to be sure that NP are completely removed from the cells. Do the Authors measured the amount of silica NP in the recovered medium? Is the amount of silica NP there the same used at the beginning of the treatment? This criticism also leads to reconsider the way to express NP doses that should be mass/surface rather than mass/volume.

- Given this possible complication, the interpretation of the so-called “recovery” should be also taken with some caution. For instance, the amount of several proteins (e.g. glycolytic enzymes) falls upon acute exposure but exhibits a clear cut restoration upon “recovery”.  Are rgwy sure that this depends upon effective silica NP removal? What happens if medium is not changed? I think that such a control should be performed to ensure a correct interpretation of the results

- Raw264.7 cells are highly glycolytic and markedly dependent by this pathway. Nevertheless, treatment toxicity seems quite modest and, more importantly, no real functional impairment is noted when LPS stimulates the macrophages. I think that this finding should be highlighted and possibly discussed.

- The methods adopted to remove possible contaminants from silica suspension, does not give the Authors the certainty of endotoxin elimination. Do the Author evaluated the level of LPS in NP suspension before treating the cells?

- In panel A, nitrite concentration is indicated in the mM range, it seems very high indeed! Moreover the difference between LPS treated and untreated cells is not too large possibly pointing to a possible LPS contamination of NP

- In the caption of figure 4, is reported the use of copper. The reviewer does not understand when and how the metal has been used.

Author Response

First and foremost, we would like to thank warmly Reviewer 3 for the very useful report that he/she sent to us. We would like to take some space to detail our responses to the specific concerns risen by Reviewer 3. Please note that in the revised version, the changes made appear in blue text color

Comment

  • When the Authors talk about “recovery from silica NP”, they simply replace the NP-additioned media with NP-free media. However, they do not consider the sedimentation of the NP on the cells. Because NP form aggregates, it is impossible to be sure that NP are completely removed from the cells. Do the Authors measured the amount of silica NP in the recovered medium? Is the amount of silica NP there the same used at the beginning of the treatment? This criticism also leads to reconsider the way to express NP doses that should be mass/surface rather than mass/volume.
  • - Given this possible complication, the interpretation of the so-called “recovery” should be also taken with some caution. For instance, the amount of several proteins (e.g. glycolytic enzymes) falls upon acute exposure but exhibits a clear cut restoration upon “recovery”.  Are rgwy sure that this depends upon effective silica NP removal? What happens if medium is not changed? I think that such a control should be performed to ensure a correct interpretation of the results

Reply

There must have been a misunderstanding here. Of course the cells still contain silica and the end of the exposure and during the recovery period, and all the question is how cell react when they do not internalize silica any further but still contain some. To make our point clearer, we have now added figure 1

Comment

- The methods adopted to remove possible contaminants from silica suspension, does not give the Authors the certainty of endotoxin elimination. Do the Author evaluated the level of LPS in NP suspension before treating the cells?

Reply

The internal control in this case is the secretion of interleukin 6. We have performed a dose response experiments, and we know from it that the levels detected with silica alone match to levels well below 1 ng/ml LPS, i.e. levels that would be undetectable by classical endotoxin detection kits. It happened to us in other research projects to be plagued by LPS contamination on NPs, this really shows at once with much higher values of secreted IL-6

Comment

- In panel A, nitrite concentration is indicated in the mM range, it seems very high indeed! Moreover the difference between LPS treated and untreated cells is not too large possibly pointing to a possible LPS contamination of NP

Reply

Thank you very much for pointing out this mistake. The scale was in µM and not in mM. This has now been corrected

Comment

- In the caption of figure 4, is reported the use of copper. The reviewer does not understand when and how the metal has been used.

Reply

That was a mixup with another paper that we are writing on macrophages and copper. Thank you for pointing out the mistake, which we corrected

Reviewer 4 Report

The manuscript aims to prove the long-term effects of an acute exposure of pyrolytic silica on murine macrophages starting from a proteomic analysis to validation with functional assays. The experimental work is well designed but the manuscript presents different and some relevant limitations. I listed my main comments below:

  1. The title highlights the proteomic approach used for this study but no proteomic data is shown in the manuscript. The authors refer to supplementary material for proteomic data. I suggest to add in the main text at least the results of pathways analysis. Moreover, this analysis was not described at all in Materials & Methods section only mentioned in the results (DAVID toll). Please, add proteomic data and analysis description.
  2. In the introduction, line 52-53, the authors mentioned unexpected results (Refs. 5 and 7) regarding omics studies of the effects of silica on cells. Which are these “unexpected results”? Please, add more information on the main results of the works cited and the pertinence with the goal of the study.
  3. The authors sterilized the suspension of silica nanoparticles by pasteurization. This method is not enough to remove the bacterial endotoxin eventually present in the NP solution. For study aimed to evaluate an inflammatory response of innate immune cells, especially monocytes and macrophages, exposed to nanoparticles, it is crucial to test and quantify the presence of LPS in nanoparticle solution by LAL test. Innate immune cells are very reactive also to minimal amount of LPS and NP can bind LPS present in the environment. Did authors measure the amount of LPS present in the solution? This aspect is important to evaluate the inflammatory response without any bias.
  4. For the characterization of silica NP the authors show PDI simply as a percentage. Did they measure the DLS intensity or number size distribution? Please, specify.
  5. Figure 3A, B, C is unacceptable. The decrease of spikes on cell surface is not evident after NP exposure and cannot be deduced only by visual observation.
  6. A prolong inflammatory effect of silica NP cannot be deduced only by measuring two inflammatory cytokines, one of which does not change. These results are too weak for supporting the conclusions. Moreover, in the figure 4 A, C and D data from cells after recovery without LPS stimulation is missing.

Author Response

First and foremost, we would like to thank warmly Reviewer 4 for the very useful report that he/she sent to us. We would like to take some space to detail our responses to the specific concerns risen by Reviewer 4. Please note that in the revised version, the changes made appear in blue text color

Comment

  1. The title highlights the proteomic approach used for this study but no proteomic data is shown in the manuscript. The authors refer to supplementary material for proteomic data. I suggest to add in the main text at least the results of pathways analysis. Moreover, this analysis was not described at all in Materials & Methods section only mentioned in the results (DAVID toll). Please, add proteomic data and analysis description.

Reply

Thank you very much for your appreciation of our proteomic work. We have followed your suggestion and have added Table 1 and Table 2 to show proteomic results in the main text.

Comment

  1. In the introduction, line 52-53, the authors mentioned unexpected results (Refs. 5 and 7) regarding omics studies of the effects of silica on cells. Which are these “unexpected results”? Please, add more information on the main results of the works cited and the pertinence with the goal of the study.

Reply

The major results of these two papers are now described in lines 58-59 and 62-63 (in blue)

Comment

  1. The authors sterilized the suspension of silica nanoparticles by pasteurization. This method is not enough to remove the bacterial endotoxin eventually present in the NP solution. For study aimed to evaluate an inflammatory response of innate immune cells, especially monocytes and macrophages, exposed to nanoparticles, it is crucial to test and quantify the presence of LPS in nanoparticle solution by LAL test. Innate immune cells are very reactive also to minimal amount of LPS and NP can bind LPS present in the environment. Did authors measure the amount of LPS present in the solution? This aspect is important to evaluate the inflammatory response without any bias.

Reply

The internal control in this case is the secretion of interleukin 6. We have performed a dose response experiments, and we know from it that the levels detected with silica alone match to levels well below 1 ng/ml LPS, i.e. levels that would be undetectable by classical endotoxin detection kits. It happened to us in other research projects to be plagued by LPS contamination on NPs, this really shows at once with much higher values of secreted IL-6

Comment

  1. For the characterization of silica NP the authors show PDI simply as a percentage. Did they measure the DLS intensity or number size distribution? Please, specify.

Reply

The measure implemented in the machine is by intensity

Comment

  1. Figure 3A, B, C is unacceptable. The decrease of spikes on cell surface is not evident after NP exposure and cannot be deduced only by visual observation.

Reply

We provide a new confocal microscopy figure, made on the same coverslips but with a different microscopy procedure, to better show our point. Furthermore, we have tried to be more precise in the description made (lines 261-262, in blue)

Comment

  1. A prolong inflammatory effect of silica NP cannot be deduced only by measuring two inflammatory cytokines, one of which does not change. These results are too weak for supporting the conclusions. Moreover, in the figure 4 A, C and D data from cells after recovery without LPS stimulation is missing.

Reply

This statement is indeed not true, both IL6 and TNF show elevated levels at the end of the recovery period. This just meant that our description was not clear and it has now been changed (lines 314-320, in blue).

In now Figure 5 A and D the data without LPS are the black bars, and the recovery datapoint is present (recov bars). In now Figure 5 B and C, the scales for IL-6 are so different without and with LPS (pg vs ng) that we had to make two panels. Here again the data at the end of the recovery period are present (recov bars)

For the significance of the data we have sought advise from a statisitician, and we have been told to replace Student T-test by Mann Whitney U test. We have done that for all targeted experiments, but it showed results only on cytokines production, and we have modified the figure accordingly.

Round 2

Reviewer 3 Report

In the present form, the manuscript complies with most of the changes requested.

In general, I think that it should clearly stated that this is an acute test and that the "removal" of amorphous silica nanoparticles from the test system may still lead to a minor amount of particles that can remain internalized, even though temporarily. As for many chemicals, the residual dose cannot be sufficient for eliciting further biological effects and interfere with cell behavior.

Even though this screening assay may be useful to assess the hazardous properties of nanomaterials, more effort should be made to assess effects resulting from repeated exposure.

Reviewer 4 Report

I found the authors' responses to my comments/suggestions quite satisfactory and the revised version has been much improved. No further comments.